# Advances in Genetic and Molecular Understanding of Alzheimer’s Disease

**DOI:** 10.3390/genes12081247

**Published:** 2021-08-15

**Authors:** Laura Ibanez, Carlos Cruchaga, Maria Victoria Fernández

**Affiliations:** 1Department of Psychiatry, Washington University School of Medicine, 660 S. Euclid Ave. B8134, St. Louis, MO 63110, USA; ibanezl@wustl.edu (L.I.); cruchagac@wustl.edu (C.C.); 2Neurogenomics and Informatics Center, Washington University School of Medicine, 660 S. Euclid Ave. B8134, St. Louis, MO 63110, USA; 3Hope Center for Neurological Disorders, Washington University School of Medicine, 660 S. Euclid Ave. B8111, St. Louis, MO 63110, USA

**Keywords:** Alzheimer disease, amyloid β, tau, APOE, TREM2, neuroinflammation, OMICS, biomarkers, therapeutics

## Abstract

Alzheimer’s disease (AD) has become a common disease of the elderly for which no cure currently exists. After over 30 years of intensive research, we have gained extensive knowledge of the genetic and molecular factors involved and their interplay in disease. These findings suggest that different subgroups of AD may exist. Not only are we starting to treat autosomal dominant cases differently from sporadic cases, but we could be observing different underlying pathological mechanisms related to the amyloid cascade hypothesis, immune dysfunction, and a tau-dependent pathology. Genetic, molecular, and, more recently, multi-omic evidence support each of these scenarios, which are highly interconnected but can also point to the different subgroups of AD. The identification of the pathologic triggers and order of events in the disease processes are key to the design of treatments and therapies. Prevention and treatment of AD cannot be attempted using a single approach; different therapeutic strategies at specific disease stages may be appropriate. For successful prevention and treatment, biomarker assays must be designed so that patients can be more accurately monitored at specific points during the course of the disease and potential treatment. In addition, to advance the development of therapeutic drugs, models that better mimic the complexity of the human brain are needed; there have been several advances in this arena. Here, we review significant, recent developments in genetics, omics, and molecular studies that have contributed to the understanding of this disease. We also discuss the implications that these contributions have on medicine.

## 1. Introduction

Ever since Alois Alzheimer provided the first clinical and pathological description of this disease in 1901, we have learned that Alzheimer’s disease (AD) is a complex and multifactorial condition in which the interplay of both genetic (65%) and lifestyle (35%) factors [1] is involved in the accumulation of protein aggregates of β-amyloid (Aβ) and tau in the brain that ultimately causes neuronal death and loss of gray matter. AD has had an estimated cost to the United States healthcare system of USD 290 billion. Disease prevalence is expected to grow from 5.8 million in 2019 to 14 million by 2050 [2]; hence, extensive international research efforts have been devoted to deciphering the causes of disease and developing therapeutics that may alter the course of the disease. Results have been elusive for several reasons.

There are three main etiological categories in AD: autosomal dominant AD (ADAD), early onset AD (EOAD), and late-onset AD (LOAD). Mutations in one of the three genes with Mendelian inheritance that cause disease (amyloid precursor protein (*APP*) and presenilin 1 and 2 (*PSEN1*, *PSEN2*)) are normally present in the ADAD form, with early onset (before 65 years old) and rapid progression. This form is fairly rare, about 1% of cases, but it has been instrumental for our initial understanding of the pathology of the disease, the development of animal models, and the design of the first therapeutic treatments. *APP*, *PSEN1,* and *PSEN2* are members of the same Aβ processing pathway. The identification of specific mutations directly related to the main pathological hallmark of AD, extracellular aggregates of Aβ plaques, led to great advances in our understanding of the disease and to the formulation of the amyloid cascade hypothesis [3]. The amyloid cascade hypothesis states that a malfunction in the system causes an accumulation of Aβ in the brain that triggers a cascade of events, ultimately resulting in cell death.

The remaining 99% of cases are largely classified into EOAD (~5%) or LOAD (~95%) according to the age of disease onset, with a threshold arbitrarily established at 65 years old. In addition, these can also be further categorized into sporadic AD (sAD) or familial AD (fAD), depending on the incidence of cases within families. Unless specified, for the remainder of the text, we will refer to the non-ADAD forms (EOAD, LOAD, sAD, and fAD) as AD. The non-ADAD forms present a more complex genetic architecture, with associations to over 29 genetic loci identified to date [4,5,6,7]. The loci identified through genetic studies have suggested alternative pathways beyond those involved in Aβ accumulation, such as tau aggregation, lipid metabolism, the innate immune response, and endosomal vesicle recycling. It is not clear whether any of these pathways have a greater role than the others. On top of this complexity, microglia are active players in the clearance of Aβ plaques whose activation seems to be regulated by APOE; yet, hyper activation of microglia is detrimental [8,9] (Figure 1).

This complexity raises questions for the “one-size-fits-all” approach. Critics of the amyloid cascade hypothesis have stated that the failure of Aβ-targeted drugs is partly due to the fact that ADAD may be different from AD. As such, a plethora of potential drug targets have been envisioned, but most have been unsuccessful for various reasons. First, it is unclear how and when the implicated genes and pathways interact and if they are “active” in all individuals. Second, a definitive diagnosis of AD cannot be made without confirmation by autopsy, so physicians and scientists have to rely on biomarkers (e.g., measuring Aβ, tau, or p-tau in cerebrospinal fluid (CSF) or plasma, or Aβ deposition in the brain using positron emission tomography (PET) imaging) to make diagnoses as accurate as possible. However, these methods are either not fully implemented (plasma), invasive (lumbar puncture for CSF), or expensive (CSF and imaging), which limits their generalized use in screening of trial participants. These diagnostic challenges lead to a “contamination” of clinical trials with non-AD cases, mostly with misdiagnosed frontotemporal dementia (FTD) cases [10] and clinically diagnosed cases that were amyloid-negative by Pittsburgh compound B (PIB) imaging or CSF ELISA (around 30% are amyloid-negative) [11]. Another major problem is that pathological changes that underlie brain degeneration and cognitive loss begin at least 10 to 20 years before dementia onset [12,13]. Most clinical trials so far have focused on individuals with clinical symptoms, in which the neurodegeneration may be too advanced for any therapeutic to reverse or stop deterioration [14]. Accordingly, current clinical trials are trying to include mild cognitive impairment (MCI) cases, defined as a transitional state between normal aging and dementia, although not all MCI patients convert to AD. Hence, a critical goal of biomedical research is to identify biomarkers of AD for these preclinical stages allowing for early diagnosis and intervention.

## 2. There Is More to Alzheimer’s Disease Than Amyloid

### 2.1. The Amyloid vs. Tau Hypotheses

The identification of mutations in the *APP*, *PSEN1,* and *PSEN2* genes in families with ADAD led to the formulation of the amyloid cascade hypothesis. The presenilin genes encode secretases (α, β, and γ) that cleave APP, a transmembrane protein, into amyloid β (Aβ) units of different lengths (from 36 to 43 amino acids in length) that are released to the extracellular space. Neurons are the main producers of Aβ, and mutations in these genes cause an overproduction of Aβ_42_ and its various toxic forms that accumulate into plaques. Plaque formation may start a series of events involving synaptic dysfunction by interfering with glutamatergic synapsis and inflammation by causing microglia hyperactivity, which promotes hyperphosphorylation of tau. Tau hyperphosphorylation can lead to the generation of destabilized microtubules in the intracellular space that aggregate and form neurofibrillary tangles (NFTs), leading to widespread neuronal dysfunction and death [15]. AD is a disease that starts with the accumulation of Aβ plaques followed by the formation of NFTs, which would be more likely to cause the observed neuronal dysfunction and degeneration [16,17] since the spreading of tau pathology is highly correlated with the patterns of clinical symptoms and cognitive decline [18]. Nonetheless, a decrease in cerebral blood flow is one of the first changes in AD pathology and could reflect dysfunction of contractile pericytes. Nortley et al. (2019) measured capillary diameters at positions near pericytes in human brain biopsies from cognitively unimpaired individuals with Aβ plaques, as well as in AD mice models (APP^NL-G-F^). They observed that capillaries were constricted near pericytes and that this constriction was correlated with the severity of Aβ deposition. In addition, oligomeric Aβ promotes the generation of reactive oxygen species (ROS) (NOX4), which triggers the release of endothelin-1, which acts on ET_A_ receptors to induce pericyte contraction. However, it is not clear what damage to synapses and neurons is due to the decrease in energy supply caused by Aβ-induced capillary constriction [19].

Another early feature of AD caused by Aβ depositions is neuronal hyperactivity. Zoo et al. (2019) demonstrated that, given a neuron-specific baseline activity driven by glutamatergic synapses, soluble Aβ blocks the reuptake of synaptically released glutamate, causing presynaptic glutamate accumulation, which increases depolarization and promotes hyperactivity [20]. Current AD treatment with memantine blocks the effects of excess glutamate that inhibits signal detection by NMDA glutamate receptors. This study suggests that targeting excitatory amino acid transporters (EAAT) may be a mechanism to therapeutically target neuronal hyperactivation at the early stages of the disease.

Immune response and inflammation are other key features in the pathology of AD (as later discussed in Section 2.3, “The underground of Alzheimer’s disease”). Upon microglia activation by Aβ deposits, the NLRP3 inflammasome assembles and initiates an inflammatory response, which contributes to the seeding and spreading of Aβ in AD mouse models [21]. Ising et al. (2019) demonstrated that in the absence of the NLRP3 inflammasome, tau hyperphosphorylation and aggregation were reduced, suggesting that tau pathology is a downstream process of the Aβ cascade and dependent on microglia activation [22].

Critics of the amyloid cascade hypothesis suggest that amyloid could be a side-effect of the disease [15] and that AD could be a disorder that is triggered by impairment of APP metabolism but progresses through tau-related pathology rather than Aβ-related pathology [16]. He et al. (2018) injected tau derived from human AD brains (AD-tau) into AD transgenic mice, overexpressing pathogenic Aβ. Mice showed accumulation of AD-tau seeds within dystrophic axons surrounding Aβ plaques; these seeds spread to neuronal somas and dendrites to recruit endogenous soluble tau and form NFTs and neuropil threads (NTs) [23]. Tau phosphorylation is mediated by neuronal p38 mitogen-activated protein kinase (p38MAPK), which is activated by Aβ plaques and cytokines (IL-1β). There are four isoforms of p38MAPK (α, β, δ, and γ), and each can phosphorylate tau at specific sites. Maphis et al. (2016) found that selective suppression of the p38αMAPK rescued late-stage tau pathology and improved working memory in 20 months old mice expressing human tau (hTau) [24]. On the other hand, Ittner et al. (2016) observed that depletion of p38γMAPK in APP23 mice increased cognitive deficits whereas increased expression of p38γMAPK (i.e., increased tau phosphorylation) abolished those deficits. In addition, they observed that APP23.p38γ^−/−^.tau^−/−^ mice did not present memory deficits, suggesting that the effects of p38γ were tau-dependent [25]. While no mutations have been found within the MAPK pathway that are associated with AD, somatic mutations in the *BRAF* gene (which is part of the MAPK pathway) in the erythro-myeloid progenitor lineage in mice may cause neurodegeneration [26].

Finally, Klein et al. (2019) studied the histone 3 lysine 9 acetylation (H3K9ac) mark in 669 aged brains from the Religious Order Study (ROS) and the Rush Memory and Aging Project (MAP) and correlated it to their Aβ and tau pathological signatures. Almost 23% of H3K9ac domains were associated with tau protein load, whereas only 2% were associated with Aβ. The tau-associated domains clustered in large genomic regions within gene promoter or enhancer regions and in open chromatin compartments. Using induced pluripotent stem cell (iPSC)-derived neurons, they further showed that overexpression of *MAPT*, without tangle formation, is enough to induce chromatin reorganization, suggesting that the tau effects in epigenomic architecture are an early event in tau pathology [27].

### 2.2. Polyvalent APOE

ApoE is a protein that transports lipids from one tissue or cell to another. It is highly expressed in the liver, adipose tissue, and artery wall, but it is also found in the central nervous system (CNS), where it is mainly synthesized by astrocytes and microglia [28]. Two SNPs (rs429358 and rs7412) within the *APOE* gene generate three major allelic variants (ε2, ε3, and ε4), which have a worldwide frequency of 8.4%, 77.9%, and 13.7%, respectively [29]. These isoforms bind to lipids, receptors, and Aβ with varying efficiencies [28,30,31,32]. The presence of the APOE ε4 allele has been associated with hyperlipidemia and hypercholesterolemia [33,34]; one copy of ε4 increases risk for AD by ~3-fold and two copies by ~12-fold [35], yet only 40% of sporadic AD and 60–70% of LOAD families carry this allele [29]. In addition, having the ε4 allele correlates with an average of 2–5 years earlier AAO, or up to 10 years if carrying two copies of the ε4 allele [36,37]. This risk not only applies to sporadic or familial LOAD but also to ADAD [38]. The ε2 allele is considered protective and would delay the appearance of symptoms [36,39,40]. The ε3 allele has a neutral effect, although rare mutations associated with this isoform (*APOE3*-Christchurch p.R136S) confer protection against developing the disease when occurring in homozygosis [41].

It has been suggested that APOE contributes to AD pathology through both Aβ-dependent and Aβ-independent pathways. In an isoform-dependent manner, free APOE can influence Aβ deposition, but it can also help soluble Aβ to cross the blood-brain barrier (BBB) [42,43,44]. Alternatively, lipidated APOE recruits soluble Aβ preventing Aβ plaque formation, but also facilitates its cell-absorption by neprilysin, produced by microglia, or by cell-surface receptors (LRP1, LDLR, and HSPG) where it is degraded at the lysosomes [45,46,47]. However, recent studies suggest that APOE secreted by glia stimulates *APP* transcription and Aβ production in neurons in an isoform-dependent manner [48].

On the other hand, *APOE* has been associated with CSF tau levels [49,50,51]. iPSC-derived neurons expressing ApoE ε4, but not ApoE ε3, had higher levels of tau phosphorylation [52]. Similarly, tau transgenic mice that express human *APOE* had higher tau levels in the brain and a greater extent of somatodendritic tau redistribution compared to *Apoe^−/−^* mice [52]. More importantly, through gene editing, Wang et al. [53] converted *ApoE* ε4 to *ApoE* ε3 and was able to rescue the normal phenotype.

Finally, beyond its effect on amyloid and tau, ApoE also influences microglial activation, the latter possibly through Trem2 interaction [54,55]. According to Krasemann et al. [55], the transition of microglia from a homeostatic- to a disease-associated microglial (DAM) phenotype would be dependent on ApoE. Supporting this, Ulrich et al. (2018) found that Apoe deficient mice presented a significant reduction in fibrillar plaque-associated microgliosis and activated microglial gene expression [8]. Recently, Parhizkar et al. (2019) showed that amyloid plaque seeding is increased in the absence of functional Trem2 and that this seeding goes along with decreased microglial clustering and reduced plaque-associated ApoE [56]. Yet, it is uncertain how Trem2 interferes with microglial lipid metabolism [57].

### 2.3. The Underground of Alzheimer’s Disease—The Immune System

Early genome-wide association studies (GWAs) were successful at identifying additional genetic risk factors for AD, such as *CLU*, *PICALM*, *CR1*, *BIN1,* and *CD33* [4,58,59,60]. The immune pathway was seen as an important component of AD pathology since *CLU*, *CR1,* and *CD33* have putative functions in the immune system. More recent studies with larger data sets identified additional genome-wide significant genes involved in the immune pathway, including *MS4A*, *CD2AP*, *EPHA1*, and *ABCA7* [61,62]. The later discovery of loss-of-function variants in *TREM2* provided scientists with particular targets to focus on in the study of the immune response in AD pathology [63,64]. More recently, it was found that the minor allele of rs1057233 (G), near the GWAS *CELF1* risk locus [4], showed an association with lower expression of *SPI1* in monocytes and macrophages [65]. *SPI1* encodes PU.1, a microglial transcription factor critical for myeloid cell development, which regulates the expression of numerous AD risk genes (*TREM2*, *TYROBP*, *CD33*, *MS4A* cluster genes, and *ABCA7*) [54,65]. Recent genome-wide meta-analyses of AD-by-proxy individuals identified 29 risk loci that are strongly expressed in immune-related tissues and cell types [6]. Two of these genes, *ADAM10* and *ACE,* along with *TREM2* and *SPI1,* were found to have a genome-wide significant association in the largest known GWAS that included around 95,000 people [7]. *ADAM10* is the α-secretase for APP that produces a secreted ectodomain fragment (sAPPα) that has neuroprotective and neurotrophic properties. In addition, ADAM10 cleaves Notch and various immune and growth factor proteins [66]. *ACE* encodes an enzyme involved in the conversion of angiotensin I into a physiologically active peptide, angiotensin II, a potent vasopressor. ACE is also involved in Aβ degradation [67]. It is still unclear how mutations in these genes relate to microglial dysfunction, but overexpression of *ACE* in microglia and macrophages in a double transgenic mice model for AD (APPswe/PS1dE9) substantially reduced cerebral soluble Aβ_42_, vascular and parenchymal Aβ deposits, and astrocytosis [68].

Activation of p38MAPK signaling in microglia (due to Aβ plaques) releases proinflammatory cytokines in astrocytes and neurons, resulting in inflammation and tau phosphorylation [69]. Deficits in *TREM2* have also been linked to dysregulation in PPARγ/p38MAPK signaling. Microglia switch from using oxidative phosphorylation for energy production to glycolysis in the presence of Aβ plaques. This metabolic reprogramming depends on the mTOR-HIF-1α pathway [70]. Piers et al. (2019) observed that iPSC-derived microglia from patients carrying pathogenic *TREM2* mutations had trouble switching to glycolytic metabolism, which ultimately was reflected by dysregulation of the PPARγ/p38MAPK signaling [71].

Microglia are also stimulated by Aβ plaques through transmembrane proteins CD33 and TREM2. While CD33 activation dampens microglial phagocytosis by inhibiting phosphatidylinositol-3 kinase (PI3K), TREM2 responds to ligand binding by activating PI3K to increase phagocytosis [72]. Functional analysis suggests that downregulation of *CD33* may be beneficial to AD since amyloid levels were reduced in a mouse model of AD (APPswe/PS1dE9) that were also *CD33*^−/−^ [73,74]. However, the consequence of regulating *TREM2* expression is unclear. For example, higher soluble TREM2 (sTREM2) in MCI or AD individuals was associated with reduced rates of cognitive decline and clinical progression [75]. *Trem2* knockout in a mouse model of tauopathy (PS19) resulted in a reduction in neurodegeneration and inflammation [76]. However, loss of Trem2 function increased the seeding and spread of neuritic plaque aggregates in mouse models of AD (APPPS1-21) injected with human AD-tau [77]. In contrast, overexpression of *TREM2* in BV-2 cells (an immortalized murine microglial cell line) promoted clearance of Aβ products and mediated neuroinflammation by downregulating the expression of inflammatory factors [78]. These apparently conflicting roles for *TREM2*, protective vs. harmful, could be due to the disease stage examined in each study [79].

Studies in 5XFAD mouse models indicate that *TREM2* is essential for microglia to acquire a DAM phenotype. However, in human AD, the DAM signature of microglia seems to be conditioned by the expression of the *IRF8* transcription factor. Loss-of-function mutations in *TREM2* promoted a less reactive phenotype of microglia [80], suggesting that the risk-effect that *TREM2* exerts on AD may be regulated by third parties. In fact, recent studies suggest that *TREM2* could be regulated indirectly through *MS4A4A*. A common variant in the MS4A cluster (rs1582763) is associated with increased CSF sTREM2. This study also demonstrated that TREM2 is implicated in disease in general and not only in those individuals that carry *TREM2* risk variants. Mendelian randomization analyses demonstrated that high sTREM2 levels were protective. In addition, it was found that MS4A4A and TREM2 co-localize intracellularly, suggesting MS4A4A as a potential therapeutic target for AD [81]; Alector, Inc. is currently testing an antibody that mimics the protective effect of the MS4A4A variant.

Finally, it seems that microglia and the autophagy pathway may interact in the pathology of AD disease. Hung et al. (2018) described deficits in the lysosome and autophagosome pathways using iPSC-derived neurons from individuals carrying pathogenic mutations in *PSEN1* and *APP* [82]. However, the disruption of these pathways seems to be more pronounced in the LOAD forms, for which several genes have been associated [83] and in relation to the expression decline of some proteins in the autophagy pathway due to age, which is exacerbated in AD [84]. Since microglia are the main cells phagocytizing Aβ plaques, Heckmann et al. (2019) hypothesized that defects in the autophagy pathway could influence microglial behavior in AD [84]. Using the 5XFAD AD mouse model, they identified that LC3 associated with endosomal membranes (LC3-associated endocytosis—LANDO) supports the clearance of Aβ deposits and prevents microglia activation. However, this process is dependent on the presence of several autophagy regulators, including ATG5, whose expression decreases with age [84].

Another gene implicated in lysosome and autophagosome dysfunction and risk for AD is *TMEM106B*. This gene has been reported to be associated with FTD in granulin (*GRN*) mutation carriers [85] and with AD interacting with *APOE*. More recently, Li et al. used a digital deconvolution [86] to estimate the brain cell-type proportion from multiple cohorts. Genetic scans of neuronal proportion indicate that a variant located in the *TMEM106B* gene is the major regulator of neuronal proportion in adults but not young individuals [87]. Impaired lysosomal function reduces lysosomal degradative efficiency, which leads to an abnormal build-up of toxic components in the cell. An impaired lysosomal system has been associated with normal aging and a broad range of neurodegenerative disorders, including AD [87]. These findings suggest that *TMEM106B* could be a potential target for neuronal protection therapies to ameliorate cognitive and functional deficits.

## 3. Unraveling the Molecular Mechanisms in AD Pathogenesis

Most of our knowledge of AD genetic risk factors originated from studying blood samples; yet, the genome is transcribed and translated differentially across tissues in response to different transcription factors, metabolic signals, and environmental responses. Accordingly, in recent years there has been an effort to study different omic layers (genome, transcriptome, proteome, metabolome, and epigenome) in different tissues affected by AD (blood, plasma, CSF, and brain) and different cell types (macrophages, neurons, microglia, astrocytes, and oligodendrocytes), whether it is in human samples, mouse models of AD, or iPSCs. These studies have advanced our understanding of the roles of amyloid, tau, APOE, and the immune system in identifying the pathological triggers and order of events in pathogenesis. Recent studies suggest that DNA is not static during an individual’s lifetime and is a feature of the aging brain. This DNA instability is worse in AD [88,89,90]. As such, the existence of somatic genetic mosaicism was suggested after detecting increased *APP* copy number variants (CNV) in cortical neuronal nuclei of sporadic AD patients [91], although this event would only contribute to a small percentage of sporadic AD cases [92].

Most of our current understanding of processes downstream of the genome comes from the analysis of the RNA (in its multiple species), whether it comes from blood, bulk brain tissue, or, more recently, from specific cell types. Using whole transcriptome profiling of AD brains, over 2000 genes were found to be deregulated in AD cases [93], and most of these were associated with functional pathways involved in the immune response, apoptosis, cell proliferation, energy metabolism, and synaptic transmission, corroborating findings from previous GWAs analyses [94,95]. Yet, this transcriptome profile may differ depending on the main AD risk factor. Network analysis of transcriptomic data from AD patients identified aging-associated processes (inflammation, oxidative stress, and metabolic pathways) were differentially altered depending on *APOE* genotype (44 vs. 33). Integration of these results with GWAs data indicated an epistatic interaction between *APOE* and several genes in the Notch pathway, suggesting a possible link between *APOE* and its role in inflammation and oxidation [96].

Integration of transcriptomic data with other phenotypes can also reveal important aspects of the disease. Transcriptome-wide network analysis with longitudinal cognitive data was used to identify a set of co-expressed genes that are related to both Aβ and cognitive decline and are separate from those that cause AD pathology [97]. Using PET imaging and brain transcriptomic data, Sepulcre et al. (2018) found an association between gene expression profiles and Aβ and tau pathology progression across the cerebral cortex. Aβ propagation was related to a dendrite-related genetic profile mostly driven by the *CLU* gene; tau propagation was related to an axon-related genetic profile led by the *MAPT* gene. This study helps to clarify the possible relationships between Aβ and tau pathology. For example, *BACE1*, the gene that codifies for the β-secretase enzyme that cleaves APP, was identified as one of the central genes in the tau-related interactome network. In addition, a lipid metabolism category was identified as commonly involved in the propagation of both Aβ and tau. APOE had a dominant role; participants who were *APOE* ε4^+^ had a linear relationship between the propagation pattern for Aβ and tau compared to those who were *APOE* ε4^−^ [98]. This suggests that a person’s genetic profile may define whether the spread of pathology is due to Aβ or tau.

Bioinformatic deconvolution approaches can untangle the transcriptomic signature of bulk brain tissue and infer the relative contribution of different cell types to a particular cell expression pattern [63]. These methods revealed that carriers of pathogenic mutations in *APP, PSEN1, PSEN2,* or *APOE* presented with lower neuron and higher astrocyte proportions compared to patients with sporadic AD, suggesting that the presence of AD genetic risk factors affects the cellular composition of AD brains [86]. Technological advancements have enabled the sequencing of individual cell nuclei, which allows for the identification of cell-specific patterns. Pioneering studies using this technology in human AD brains were capable of identifying cell-type-specific transcriptomic profiles [99]. This technology also allows the mapping of specific cell profiles at certain points in time. It was found that early in the pathology, the disease-associated transcriptional changes were highly cell-type-specific, whereas, in later disease stages, the transcriptional signature of the disease was common across cell types, mostly centered around global stress response [100]. Similarly, these technologies reveal that human microglia have an AD-related gene signature that is distinct from that described in mouse models [101], suggesting that mouse models of AD may not be adequate in vivo systems to study all pathological aspects of the disease, as discussed later in this section.

Circular RNA (circRNA) are formed by back-splicing (head-to-tail splicing) of messenger RNAs during normal processing. They were first described in eukaryotic cells, and later studies suggested that they were enriched in the synapse, acting as sponges of micro RNA (miRNA). One of the events implicated in the pathophysiology of AD is synapse loss [102]. CircRNAs were also found to be co-expressed with known causal AD genes, such as *APP* and *PSEN1*, suggesting that some circRNA are also part of the causal AD pathway. CircRNA brain expression explained more about AD clinical manifestations than the number of *APOE* ε4 alleles, suggesting that cirRNA could be used as biomarkers for AD.

Epigenetics can lead to changes that affect gene activity and expression but do not require changes of the nucleotide sequence. The main epigenetic marks are DNA methylation and histone modifications. Pioneering epigenome-wide association studies (EWAs) of AD examined the hypermethylation of CpG sites in the brain cortex of AD patients [103,104]. These studies identified methylation changes in the *ANK1* gene. Cell-specific EWAs of neuron and glia single nuclei validated and assigned methylation of *ANK1* as specific to glia [105]. More recently, Smith et al. (2019) performed a targeted methylation analysis finding that differential *ANK1* methylation is a common feature across the entorhinal brain cortex of subjects with AD, Huntington’s disease (HD), or Parkinson’s disease (PD), but not those with vascular dementia (VD) or dementia with Lewy bodies (DLB) unless these individuals had co-existing AD pathology [106]. Other studies have looked at histone acetylation marks, H4K16ac particularly, which is an epigenetic modification of the DNA that serves to regulate chromatin compaction, gene expression, stress responses, and DNA damage repair. H4K16ac marks are usually enriched with aging, but the exploration of brain temporal lobe tissue from AD patients revealed losses of acetylated histone H4K16ac, which was superior in the proximity of genes linked to aging and with previously identified AD genetic loci [107].

Metabolic decline is one of the earliest symptoms detected in patients with MCI [108]. Hence, by identifying those metabolites that differ between MCI to AD patients, it is possible to establish panels of time-specific metabolic biomarkers, which will help us understand the mechanisms of disease at different stages. Several metabolites, such as alanine, aspartate, and glutamate, have been associated with AD and cognitive decline, whereas unsaturated fatty acids have been associated with early memory impairment [109,110,111]. As such, the Alzheimer’s Disease Metabolomics Consortium observed that preclinical AD cases were enriched in sphingomyelins and ether-containing phosphatidylcholines compared to symptomatic AD cases in which acylcarnitines and several amines were the most representative metabolomic groups [112]. Similarly, sphingolipids were found to be the more distinct species between AD cases and controls and were associated with the severity of AD pathology at autopsy and AD progression [113,114]. The correlation between these metabolic signatures in the brain and peripheral tissues, as well as their relationship with key AD pathological biomarkers, remains to be elucidated, but they are a promising source of novel biomarkers.

Despite the potential of these novel technologies, a major challenge in advancing this line of research is access to sufficient brain tissue from different stages of the disease to explore pathology at different time points. iPSCs have become useful models to study single-cell behavior in disease [115]. Similarly, monocyte-derived microglia-like (MDMi) cells recapitulate key aspects of microglia phenotype and function, and their expression of neurodegenerative disease-related genes is different from that of monocytes [116]. These models have been useful for studying the effects of specific variants on cell phenotype, e.g., the study of ADAD mutations [117] and the effects of tau-related mutations in AD [118]. Most importantly, these studies help differentiate functional responses observed in mice from those in human systems [119]. Transcriptomic analysis of 5XFAD mice and human AD single nuclei brain cells revealed discordances in the transcriptomic signature of oligodendrocytes, astrocytes, and microglia between these two systems [80].

Yet, the brain is a complex organ involving the interaction of multiple cell types, with different proportions in different brain areas. In addition, AD pathogenesis is a combination of Aβ accumulation, phosphorylated tau (p-tau) formation, hyperactivation of glial cells, and neuronal loss. Therefore, iPSC or MDMi alone cannot be expected to model the brain response to AD pathogenic events. A novel engineered model has been developed that grows three-dimensionally interacting neurons, astrocytes, and microglia in order to model AD pathogenesis [120]. This new 3D human AD triculture system mirrored the first pathogenic AD stages, Aβ aggregation and p-tau formation, and the induction of microglia recruitment that leads to marked neuron and astrocyte loss [120]. More recently, several groups have managed to incorporate the microvasculature into these organoids, providing them with blood-brain barrier characteristics [121]. These models have the potential to advance our understanding of AD in multiple ways. First, we can study the pathological processes that occur in the brains of AD patients. By combining patient-specific iPSC with triculture 3D technology, we could evaluate the differential activation of pathways in a patient-specific manner. Subsequently, different drugs could be tested to evaluate the efficacy and occurrence of side effects in a patient-specific and personalized manner.

## 4. Early Prediction and Diagnosis Are Key to Better Treatment

Current diagnostic tools for AD patients include neuropsychological tests to assess memory and other cognitive abilities, whether it is Clinical Dementia Rating (CDR) [122], Mini-Mental State Exam (MMSE) [123], Montreal Cognitive Assessment (MoCA) [124], or Consortium to Establish a Registry for Alzheimer’s Disease (CERAD) [125] and measurement of biomarkers in the brain, CSF, or blood. Biomarkers can be measured in the brain using imaging techniques (MRI, CT, or PET) that inform us about metabolic changes in the brain (glucose) or deposit of certain protein aggregates (Aβ, tau, and p-tau) or in biofluids (blood and CSF). A definitive diagnosis of AD can only be made by pathological exam of the brain postmortem. However, it is known that pathologic changes in the brain occur years before a person starts showing signs of cognitive impairment [126]. A key factor in the success of clinical trials and in the treatment of AD patients is the ability to intervene before clinical symptoms appear. Thus, we are in urgent need of tools that can identify individuals at risk and be applied at the population level in a fast and affordable manner. The challenge remains in having access to well-characterized, large cohorts with a longitudinal repository of body fluids in which sets of biomarkers can be investigated in a retrospective manner. In addition, as we have reviewed in this manuscript, not only Aβ and tau contribute to the pathogenesis of AD; there are many other pathways involved that have the potential to provide accurate biomarkers to predict and follow disease progression and response to potential treatments. Clinical trials for drug targets have mainly focused on reducing the production of Aβ or trying to clear its deposits from the brain, mainly based on the knowledge derived from ADAD patients. Yet, the pathological mechanisms starting and driving ADAD, EOAD, and LOAD might differ, thus, different therapeutic approaches that take into account the etiology and genetic background of each individual should be investigated. In the next section we summarize the most recent papers on risk prediction, detection, diagnosis, and treatment strategies (Table 1).

### 4.1. Risk Prediction and Prevention

Polygenic risk scores (PRS) aim to generate a genetic profile of an individual and to predict this individual’s chances of developing a certain condition. The first PRS for AD included all 21 variants identified by the IGAP consortia with a prediction accuracy of 78.2% [135], although its accuracy could reach 82% depending on the AD subtype [136]. In fact, PRS calculated for stratified AD etiologies revealed an accuracy of 75% for fAD, 72% for sAD, and 67% for AD [127]. Under the hypothesis that a pathway-specific PRS could be more powerful at predicting certain pathological aspects of the disease, Darst et al. (2016) clustered the SNPs for the AD PRS into the major AD pathways (Aβ clearance, cholesterol metabolism, and immune response) and tested their association with cognition function and AD-biomarkers (Aβ imaging, CSF Aβ, tau, and p-tau) [128]. Unfortunately, these prediction values were no more accurate than models including all known disease variants, suggesting there is room for improvement of these predictors. Kauppi et al. (2018) generated a PRS that predicted progression from MCI to AD over 120 months; when these data were combined with baseline brain atrophy score and/or MMSE score, the prediction model was significantly improved (AUC = 84%) compared to the use of the PRS alone [129]. Furthermore, adding biomarker data from CSF and imaging measurements, the same group found that individuals with high PRS and with amyloid and tau pathology showed a faster rate of cognitive decline, even among *APOE* ε4 non-carriers [130]. In a similar study, it was found that adding imaging information of Aβ and tau deposition (PET) and neurodegeneration (MRI) to a model that already included clinical and genetic information improved the prediction accuracy of memory decline [137]. Despite the increase in predictive ability, it is uncertain whether these improvements will be clinically relevant for the daily practice of predicting people at risk. We are still in need of predictors that do not rely on biomarkers of already occurring pathology.

### 4.2. Early Detection, Diagnosis, and Prognosis

In the search for biomarkers that can reflect what is occurring in the brain, CSF has been the fluid of preference given its direct contact with the CNS. Aβ peptides and tau were among the first proteins to be investigated in CSF for detection and diagnostic purposes. This was followed by the investigation of VILIP-1 (marker of neuronal injury), YKL40 (marker of inflammation), neurogranin (NGRN—marker of synaptic function), and CLU (an apoliprotein involved in several Aβ processes and a risk factor for AD). However, these markers are not disease-specific (VILIP-1, YKL40 [138], NGRN [139]) or do not present differently between cases and controls (e.g., CLU [140]). Therefore, there is still a need to identify biomarkers that (i) are AD-specific and can predict the onset of cognitive decline and (ii) are independent of Aβ and tau metabolism so that disease progression can be monitored in patients enrolled in clinical trials using drugs targeting Aβ or tau. Recently, progress has been made in the analysis of Aβ in blood (plasma) in order to replace screening in CSF and reduce its invasiveness and related expenses [75].

Soluble TREM2 (sTREM2) is detectable in CSF and serum. Its levels are elevated in MCI-AD compared to AD or controls and correlate with those of tau and p-tau, but not Aβ [132,141]. CSF sTREM2 levels increase before the onset of symptoms, but after amyloidosis and neuronal injury have already begun [132,142,143], suggesting that *TREM2* may play a critical role in the onset and progression of tau pathology and microglia activation. In addition, a higher ratio of CSF sTREM2 to CSF p-tau181 concentrations predicted slower conversion from cognitively normal to symptomatic stages or from MCI to AD dementia [75]. sTREM2 can be generated by the proteolytic action of ADAM10 (an α-secretase also involved in the cleavage of APP), but missense mutations in the immunoglobulin-like domain and stalk region have been found to interfere with the cleavage site and shedding of sTREM in opposite directions [144]. In addition, there are three alternative transcripts for *TREM2*. One lacks the transmembrane domain and encodes only the sTREM2 form. Using bulk brain transcriptomic data from AD cases, TREM2 carriers, and controls. del-Aguila et al. (2019) showed that up to 25% of sTREM2 may be translated from *TREM2* isoforms that lack the transmembrane domain; in addition, the expression of this particular isoform was significantly different in cases compared to controls [145]. The role of sTREM2 in the cascade of pathologic events remains unclear, and because of the lack of selective inflammatory markers, it is uncertain whether inflammation and microglial activation or tau-related abnormalities occur first. Yet again, it may be that the order of pathologic events may differ between ADAD cases and AD patients that do not carry mutations on those genes [146].

Neurofilament light chain (NfL) is an intrinsic protein of the axonal cytoskeleton that is released when neurons die. NfL was found in high concentrations in CSF and blood among participants of the Dominantly Inherited Alzheimer’s Network (DIAN) ~6.8 years before the onset of symptoms [133]. NfL is not an AD-exclusive biomarker, but since it is Aβ- and tau-independent, it has the potential to be used as a proximity marker and as a marker to monitor therapy response.

The contribution of neurovascular dysfunction and blood-brain barrier (BBB) breakdown to cognitive impairment is widely recognized. These both develop early in AD; however, the relationship between vascular pathology and Aβ and tau is still unknown. Nation et al. (2019) studied the CSF from cognitively normal individuals as well as from individuals with early cognitive dysfunction who were CSF Aβ+, Aβ−, p-tau+, or p-tau−. The soluble platelet-derived growth factor receptor-β (sPDGFRβ) is mainly expressed by brain vascular mural cells but not by other cells of the CNS. Nation et al. (2019) found that sPDGFRβ was increased in the CSF of individuals with advanced CDR regardless of CSF Aβ of p-tau status, suggesting that biomarkers focused on the integrity of the brain vasculature could be a novel source for biomarkers of cognitive dysfunction in both individuals with and without AD [134].

Although most blood- and CSF-based biomarkers focus on protein levels, cell-free nucleotides are also being investigated for use in diagnostic tests. Disease-specific cell-free RNA transcripts have been found at increased levels in the blood of affected individuals [147,148]. Additionally, small non-coding miRNA were used to differentiate cases from controls across different neurodegenerative diseases: AD, FTL, and ALS were differentiated from each other with accuracy ranging from 0.77 to 0.93 [149]. These studies employed samples from symptomatic patients only, so further studies are required in preclinical individuals to confirm the potential of this miRNA-based approach as a diagnostic tool.

### 4.3. Treatment

The majority of ongoing phase III clinical trials were developed under the umbrella of the amyloid hypothesis and so are mainly focused on stopping the production of Aβ or aimed at clearing Aβ plaques. A set of drugs have been designed to target the γ-secretase complex with the aim to prevent the production of Aβ altogether. However, γ-secretase cleaves not only APP but also up to another 50 transmembrane protein substrates, including Notch receptors [150]. Recent studies have revealed that ADAD mutations destabilize the intermediate enzyme-substrate complexes between APP and γ-secretase, promoting early disassociation of γ-secretase from Aβ and thereby releasing longer and more amyloidogenic Aβ peptides [151]. The second wave of current clinical trials use monoclonal antibodies to promote Aβ clearance, have been designed for a very specific subset of the population, and try to tackle the disease before symptoms appear. The Dominantly Inherited Alzheimer’s Network Trials Unit (DIAN-TU) selects participants from families with autosomal dominant mutations in either *APP*, *PSEN1,* or *PSEN2* genes and treats them with either Solanezumab, a monoclonal antibody that targets soluble Aβ, or Gantenerumab, a monoclonal antibody that interacts with Aβ plaques and activates microglia phagocytosis [152]. Similarly, the Alzheimer’s Prevention Initiative (API) ADAD trial is focused on a large Colombian family with ADAD due to a pathogenic mutation in *PSEN1* (p.E280A); members of this family are treated with Crenezumab, a monoclonal antibody that recognizes multiple Aβ forms and stimulates amyloid phagocytosis while limiting inflammation [153]. The API also has two more trials: (i) CAD106, a vaccine that combines multiple Aβ forms and aims to produce a strong antibody response while avoiding inflammatory T cell activation, and (ii) Umibecestat, which seeks reduction Aβ production by inhibiting the BACE1 protease. These trials are conducted with 60–75-year-old cognitively normal *APOE* ε4 homozygotes and aim to prevent the appearance of disease [154]. However, the predictions for this trial are not very promising, since Verubecestat, another drug aimed at inhibiting BACE1 to block Aβ production, failed to improve the cognitive abilities of prodromal AD cases [155]. That is not the case for aducanumab (Aduhelm), a monoclonal antibody against aggregated forms of Aβ approved by the FDA in June 2021 (https://www.fda.gov/drugs/news-events-human-drugs/fdas-decision-approve-new-treatment-alzheimers-disease (accessed on 3 August 2021)). Even though it has proved effective in reducing the burden of Aβ plaques, it is still not clear if it also reduces the symptomatology [156].

Despite some anti-Aβ therapeutic drugs look promising in phase III clinical trials, recent data suggest that amyloid would be aside-effect of the brain’s response to stress in sporadic AD, not a causative factor as in familial AD [157]. Cognitive decline and pathogenic events are directly associated with the initiation of tau aggregation, hence an interest in developing tau-related therapies [158]. Tau pathology in AD is characterized by a disruption of 3R to 4R tau isoforms, resulting in an approximately 2:1 4R:3R ratio [159,160]. Tau expression could be reduced with small interfering RNA (siRNA) [161] or antisense oligonucleotides (ASO) [162]. These mechanisms have not yet been tested in clinical trials for AD or other tauopathies, but they have been used for cancer [163] and spinal muscular atrophy [164].

*APOE* polymorphisms have been recognized to contribute to AD pathology by both gain-of- and loss-of-function properties. This bi-directional effect must be taken into account when designing therapies targeting ApoE [165]. On the one hand, mechanisms that enhance ApoE quantity have been shown to promote Aβ clearance and synaptic function in an isoform-dependent manner in murine models [166]. On the other hand, reduction in ApoE levels in mice models using anti-Apoe ε4 monoclonal antibody seemed to prevent cognitive impairment and brain hyperphosphorylation [167]. Recently, it was found that an anti-human ApoE antibody specifically recognizes human ApoE ε4 and ApoE ε3 and preferentially binds nonlipidated, aggregated ApoE in mouse models expressing human ApoE and human Aβ [168]. Other therapeutic approaches look at modifying ApoE properties through structural modification, an increase in ApoE lipidation, or blocking its interaction. CRISPR/Cas9 has been used to transform ApoE ε4 into ApoE ε3 in mouse astrocytes [169,170]. Recently, Wang et al. (2018) was successful at converting ApoE ε4 to ApoE ε3 in iPSC-derived neurons and proved that the introduction of ApoE ε4 recapitulated the pathogenic effects [53].

Given the importance of the immune response in the pathology of AD, therapies targeting this process, mostly through *CD33* and *TREM2*, are moving into the clinical trial phase, as announced at the 14th International Conference on Alzheimer’s and Parkinson’s Diseases, held 27–31 March in Lisbon, Portugal. In particular, two groups, the biotech Alector, Inc. and the German Center for Neurodegenerative Diseases in Munich, have developed antibodies that activate TREM2. These antibodies will trigger signaling through its co-receptor DAP12 resulting in phosphorylation of Syk and the downstream activation of microglia to remove amyloid. The Alector antibody (AL002) has moved into phase I clinical study. Similarly, Alector has started its clinical trial of the anti-CD33 antibody (AL003). Taking into account the time-specific protective vs. harmful effect that microglia have in AD, if these antibodies work as expected, they would need to be administered at very specific time points.

Finally, other features of age-related diseases are BBB integrity and the accumulation of senescent cells. BBB integrity is essential for the (i) Aβ-clearance and (ii) lipid transport. Docosahexaenoid acid (DHA) is a blood-based essential fatty acid for cognition, and current clinical trials are looking at the cognitive benefits of taking DHA diet supplements. Pan et al. (2016) showed reduced DHA levels and cognitive response in fatty acid-binding protein 5 (FABP5) knockout mice, suggesting that FABP5 upregulation could be an alternative approach to improve DHA uptake and rescue cognitive function [171]. Zang et al. (2019) studied the brains of patients with AD and the transgenic APP/PS1 mouse model of AD. They observed that oligodendrocyte progenitor cells (OPC—brain cells mobilized in response to neuronal injury and demyelination) accumulate around Aβ plaques and acquire a senescent phenotype characterized by the upregulation of p21/CDKN1A and p16/INK4/CDKN2A proteins and β-galactosidase activity. They observed that senolytic treatment (dasatinib plus quercetin) improved APP/PS1 AD mouse model condition by removing p16-expressing OPCs from Aβ plaques (after 9 days of treatment), reducing Aβ-plaque-associated proinflammatory cytokines and microglial activation, and reducing levels of inflammation and Aβ plaque size (after 11 weeks of treatment). Altogether, senolytic treatment improved the hippocampus-dependent learning and memory capabilities of APP/PS1 AD mice [172]. Quercetin is a flavonoid with antioxidant and anti-inflammatory effects found in many plants and foods such as berries, green tea, and Ginko biloba, among others; natural products have the benefit of being readily available, as such some of them are being tested in animal models, for their neuroprotective, anti-inflammatory, antioxidant, anti-amyloidogenic, anticholinesterase properties, as potential therapeutics for AD [173,174].

## 5. Conclusions and Future Directions

The recent studies of the molecular mechanisms of AD have shown us that amyloid accumulation does not only trigger tau hyperphosphorylation and immune response, but it starts other series of events that contribute to increased stress in the brain—e.g., reduction in brain blood flow or increment of neuronal hyperactivity. In addition, Aβ activates the inflammasome and p38MAPK pathway, which stimulates the production of cytokines that promote tau hyperphosphorylation. Once the pathological environment is started, *APOE* can exacerbate the situation, both through the Aβ and tau pathways in an isoform-dependent manner, but *APOE,* in turn, seems to be regulated by *TREM2*. All of the different molecular mechanisms are highly interconnected and participate in AD pathogenesis at different time points. Therefore, future research should focus on identifying the potential triggers for non-ADAD etiologies, whether it is searching for additional rare coding variants in the many loci associated with the disease or exploring the non-coding regions of the genome for downstream effects modifying gene expression. Additionally, improvements are needed in the in vitro systems used to study the disease since we have seen that the response in mouse models differs from that of humans. By better understanding the chain of pathologic events associated with different genetic risk factors, we can potentially identify AD subtypes related to specific genetic architectures allowing for personalized diagnosis and treatments. So far, our capacity to predict AD is quite limited, with PRS having a prediction accuracy between 65% and 75%. Our tools for early detection are limited as well since we cannot currently detect individuals at risk until their Aβ and tau load has already built up. Ultimately, this is a detriment to developing and testing novel therapies in the right groups of participants.

The progress we have made in recent years in the understanding of AD has been monumental. Yet, there is still substantial work to do before we fully understand and control this disease. The generation of larger genetic studies and incorporation of rare variants in prediction models will facilitate the development of improved PRS for the prediction of the baseline risk of developing AD and will also allow for the identification of potential AD subtypes. In addition, the discovery of dynamic biomarkers will enable the prediction of age at onset and the rate of progression of the disease. Omic approaches can facilitate progress in this area by exploring changes in the proteomic and metabolomic profiles of individuals at different time points. Finally, to improve and reach a personalized medicine for AD, future studies need to incorporate ethnic diversity in the recruitment process as modeling of this disease has, so far, been almost exclusively done with European and American populations of Caucasian background.

## Figures and Tables

**Figure 1 genes-12-01247-f001:**
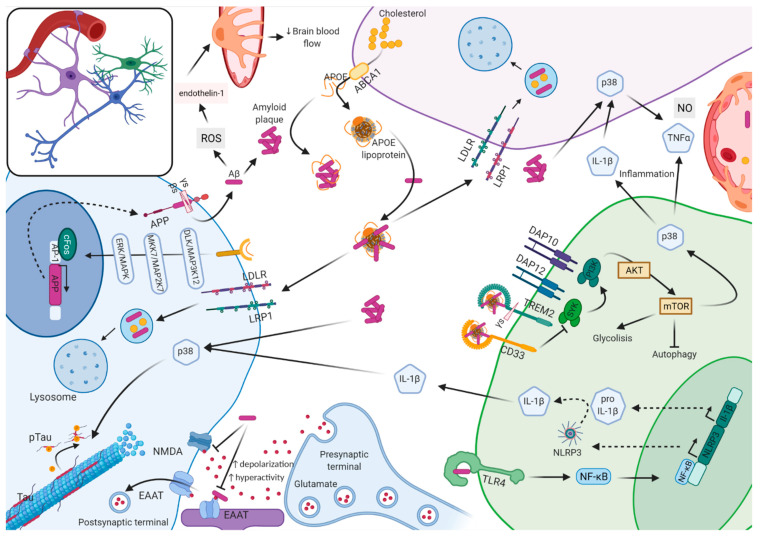
Schematic representation of the molecular interplay between neurons, astrocytes, microglia, and vasculature system in Alzheimer’s disease. Neurons produce the transmembrane amyloid precursor protein (APP). APP is cleaved by γ (γs) and β secretase (βs) into amyloid β (Aβ) units of different length aggregates extracellularly into plaques. Oligomeric Aβ promotes the generation of ROS that triggers the release of endothelin-1, causing perycite constriction, which decreases brain blood flow. Soluble Aβ blocks the reuptake of synaptically released glutamate by either N-methyl-D-aspartate receptor (NMDA) or by excitatory amino acid transporter (EAAT) receptors causing glutamate accumulation perisynaptically (excitotoxicity), which increases depolarization and promotes hyperactivity. In microglia, Aβ binds to the toll-like receptor 4 (TLR4), which causes translocation of the nuclear factor kappa-light-chain-enhancer of activated B cells (NF-κB) from the cytosol to the nucleus, where it increases the transcription of NLRP3 and pro-IL-1β. In the cytoplasm, via activated caspase-1, the inflammasome promotes the maturation of IL-1β. Amyloid plaques stimulate the activation of p38MAPK (p38) in microglia, astrocyte, and neuron. In microglia, p38 activation results in upregulation of proinflammatory cytokines, interleukin 1β (IL-1β) and tumor necrosis factor α (TNFα); IL-1β in turns activates p38 in astrocytes and neurons. In astrocytes, p38 activation causes increased expression of TNFα and nitric oxide (NO) and excitotoxicity. In neurons, p38 activation results in tau phosphorylation (Tau → p-Tau) and microtubule disassembly. APOE is mostly generated by astrocytes; free APOE can facilitate Aβ blood bran barrier (BBB) transit, but it can also accelerate aggregation and deposition of Aβ in an isoform-dependent manner. APOE can be lipidated by ABCA1 transporter-forming lipoprotein particles that bind soluble Aβ, which are then uptaken by neurons and glia via cell-surface receptors, including low-density lipoprotein receptor (LDLR) and low-density lipoprotein receptor-related protein (LRP1), and degraded at the lysosome. When free APOE binds to ApoE receptors in neurons, it can activate a non-canonical MAPK pathway, in an isoform-dependent manner, that induces cFOS phosphorylation stimulating the transcription factor AP-1, which in turn enhances transcription of APP. The complex amyloid plaque lipidated APOE can stimulate microglia through transmembrane proteins triggering receptor expressed on myeloid cells 2 (TREM2) and sialic acid-binding Ig-like lectin 3 (CD33). TREM2 activation induces the association of TREM2 to DAP12, which gets phosphorylated and recruits spleen tyrosine kinase (SYK), which activates phosphoinositide 3-kinase (PI3K) that depends on DAP10. PI3K targets protein kinase B (AKT) and activates the mammalian target of rapamycin (mTOR), which activates glycolysis, the p38MAPK pathway, and inhibits autophagy. Instead, CD33 activation inhibits PI3K. The complement receptor 1 (CR1) is a receptor for the complement components C3b and C4b and promotes the phagocytosis of Aβ.

**Table 1 genes-12-01247-t001:** Summary of selected recent studies of risk prediction, early diagnosis, and treatment.

Approaches to Risk Prediction							
Ref.		Approach		Findings			
[127]		PRS		EOAD, sLOAD, and fLOAD have different PRS profiles			
[128]		PRS + biomarker		Prediction of conversion or AAO			
[129]		PRS + brain atrophy + MMSE score		Better progression prediction			
[130]		PRS + brain atrophy + MMSE score + CSF data		Individuals with high PRS and with amyloid and tau pathology showed a faster rate of memory decline, even among APOE ε4 non-carriers			
[131]		PRS		PRS differentiate AD, FTD, PD, and ALS			
**Biomarkers for Early Diagnosis**							
**Ref.**	**Target**	**Localization**		**Indicative Of**		**Aβ-Independent**	**Tau-Independent**
[132]	sTREM2	CSF, plasma		Onset and progression of tau pathology		Y	N
[133]	Nfl	plasma, CSF		Cytoskeleton protein released with neuron death		Y	Y
[134]	sPDGRβ	blood		Blood-brain barrier breakdown		Y	Y
**Possible Treatments and Clinical Trials**							
**Clinical Trial Ref.**	**Target**	**Mechanism**	**Participants**	**Goal**	**Drug**	**Status**	
NCT01677572	Amyloid β	Monoclonal antibodies	Mild AD and MCI	Clearance of Aβ plaques	Aducanumab	Approved to treat Alzheimer’s disease	
NCT02760602			Prodromal AD		Solanezumab	T—no evidence that prodromal AD benefits from drug	
NCT02008357			Older Individuals at risk (APOE4+)		Solanezumab	P3—not recruiting	
NCT01760005			DIAN-TU		Solanezumab	P3; H—no change in cognitive performance	
NCT01760005			DIAN-TU		Gantereumab	P3; H—no change in cognitive performance	
NCT01998841			Colombian family		Crenezumab	P3—not recruiting	
NCT01661673	Y-secretase	y-modulator	Mild Cognitive Impairment	Decrease production of toxic Aβ	EVP-0962	P2—completed	

PRS—polygenic risk score; Nfl—Neurofilament Light; T—terminated; H—halted; P2—phase II; P3—phase III.

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
