# Peer review of "Advances in Genetic and Molecular Understanding of Alzheimer’s Disease"

_genes, 2021, doi:10.3390/genes12081247_

Round 1
Reviewer 1 Report
This is a nicely written review, covering the most updated information about Alzheimer's disease, especially early diagnosis and treatment. Most of the updated and relevant papers have been reviewed. While I do have the following minor suggestions:
Please consider breaking the bulky paragraphs into smaller paragraphs, e.g. section 2.1-2.3, and section 3.
Line 59 ...plaque (Fig 1). In addition, Fig 1 the resolution should be improved.
Line 73, is [?] a missing reference?
For '2. There is more to Alzheimer’s disease than amyloid', I suggest adding a paragraph talking about altered blood-brain barrier, e.g. Ab clearance, and more importantly fatty acid transport and etc (e.g. Pan et al J Neurosci 2016). Altered fatty acid transport into the brain is an important aspect to AD pathology.
Author Response
This is a nicely written review, covering the most updated information about Alzheimer's disease, especially early diagnosis and treatment. Most of the updated and relevant papers have been reviewed. While I do have the following minor suggestions:
Please consider breaking the bulky paragraphs into smaller paragraphs, e.g. section 2.1-2.3, and section 3.
Thank you for the suggestion. We have reviewed all the text to make it less bulky and more clear for a better understanding.
Line 59 ...plaque (Fig 1). In addition, Fig 1 the resolution should be improved.
We have generated a high quality figure for publication.
Line 73, is [?] a missing reference?
Yes, thanks for catching that. We have fixed this error.
For '2. There is more to Alzheimer’s disease than amyloid', I suggest adding a paragraph talking about altered blood-brain barrier, e.g. Ab clearance, and more importantly fatty acid transport and etc (e.g. Pan et al J Neurosci 2016). Altered fatty acid transport into the brain is an important aspect to AD pathology.
The reviewer is right; the integrity of the blood brain barrier is an important aspect of cognitive impairment. We already have a section in which we discuss the contribution of neurovascular dysfunction to the BBB (within Early detection, diagnosis and prognosis). As such, we find more adequate to incorporate the contribution of fatty acids to BBB in the same section. This has been incorporated as follows:
“Finally, other features of age-related diseases are BBB integrity and the accumulation of senescent cells. BBB integrity is essential for the (i) Aβ-clearance and (ii) lipid transport. Docosahexaenoid acid (DHA) is a blood-based essential fatty acid for cognition and current clinical trials are looking at the cognitive benefits of taking DHA diet supplements. Pan et al. (2016) showed reduced DHA levels and cognitive response in fatty acid-binding protein 5 (FABP5) knockout mice suggesting that FABP5 upregulation could be an alternative approach to improve DHA uptake and rescue cognitive function [171]. ”

Reviewer 2 Report
The authors have made a sincere attempt to review the developments on the fast-changing topic of Alzheimer's Disease. One of the area which needs attention is that it lacks the focus. There are long-drawn discussions on socio-economic effects, types of ADRD, causes of AD-like abeta42, tau, ApoE. It can be concised so that t reads well. It is good to provide some illustrations.
Another issue is that title suggests that the focus will be its implications on medicine. The area focusing on cure and therapeutic target(s) is somewhere in the end.
I would suggest "Revision" in trying to make it more reader-friendly. The review should be shortened, provide some illustrations about abeta, tau, ApoE, add some tables about drugs and their targets.
I would suggest the author to discuss the newly FDA-approved drug against Amyloid plaques. Aduhelm, a newly approved drug by FDA, represents a first-of-its-kind treatment approved for Alzheimer’s disease. It is the first new treatment approved for Alzheimer’s since 2003 and is the first therapy that targets the fundamental pathophysiology of the disease. Patients receiving the treatment had a significant dose-and time-dependent reduction of amyloid-beta plaque, while patients in the control arm of the studies had no reduction of amyloid-beta plaque.
Secondly, there are two schools of thought (i) believe in the amyloid hypothesis and others (ii) which do not. There should be a balanced discussion in a review and should not be skewed or biased.
Thirdly, there are issues with references. I think authors should spend some time checking the references.
Lastly, I would like you to talk about plant-based products used as potential targets for AD.
Some references that should be added:
1. Alzheimer's disease: the silver tsunami of the 21 st century
https://www.ncbi.nlm.nih.gov/pmc/articles/PMC4904444/
2. Exploring the efficacy of natural products in alleviating Alzheimer’s disease
P Deshpande, N Gogia, A Singh Neural regeneration research 14 (8), 1321
Author Response
The authors have made a sincere attempt to review the developments on the fast-changing topic of Alzheimer's Disease. One of the area which needs attention is that it lacks the focus. There are long-drawn discussions on socio-economic effects, types of ADRD, causes of AD-like abeta42, tau, ApoE. It can be concised so that t reads well. It is good to provide some illustrations.
Following suggestions from Reviewer 1 and Reviewer 2 we have broken up and simplified some of the paragraphs to make them less bulky and so that it reds better. Yet, some of that discussion has to remain to give context into the need of Precision medicine and the failure of some drugs and trials. We hope the new version is easier to follow. Regarding Illustrations, we provide Figure 1 that summarizes the interaction of all molecular players we discuss through section.
Another issue is that title suggests that the focus will be its implications on medicine. The area focusing on cure and therapeutic target(s) is somewhere in the end.
We agree, we have changed the title accordingly so it is not misleading
I would suggest "Revision" in trying to make it more reader-friendly. The review should be shortened, provide some illustrations about abeta, tau, ApoE, add some tables about drugs and their targets.
Figure 1 includes a graphical representation of all the risk factors and pathways discussed in this manuscript. We have not included detailed illustrations of Aβ, Tau and ApoE since those can be found in many other revisions. Similarly, in Table 1 we have provided a summary of the therapeutic drugs in clinical trials discussed in the manuscript that details the target, mechanism, trial participants, goal of the drug and status of the trial. Through the text, we refer the reader to more detailed reviews on treatments under study for each of the major AD targets: Aβ, Tau and APOE.
I would suggest the author to discuss the newly FDA-approved drug against Amyloid plaques. Aduhelm, a newly approved drug by FDA, represents a first-of-its-kind treatment approved for Alzheimer’s disease. It is the first new treatment approved for Alzheimer’s since 2003 and is the first therapy that targets the fundamental pathophysiology of the disease. Patients receiving the treatment had a significant dose-and time-dependent reduction of amyloid-beta plaque, while patients in the control arm of the studies had no reduction of amyloid-beta plaque.
We agree with the reviewer. At the time of writing and submission that drug was not approved yet. We have included the recent events to the review so it is up to date.
“That is not the case for aducanumab (Aduhelm), a monoclonal antibody against aggregated forms of Aβ approved by the FDA in June 2021. Even though it has proved effective in reducing the burden of Aβ plaques, it is still not clear if it also reduces the symptomatology [156].”
Secondly, there are two schools of thought (i) believe in the amyloid hypothesis and others (ii) which do not. There should be a balanced discussion in a review and should not be skewed or biased.
The reviewer is right, the filed in divided between those that believe in amyloid as being the origin of the disease vs those that think that Tau is the cause of the disease. We actually have dedicated a whole section to this issue, entitled “1.1. The amyloid vs Tau hypotheses”. Also, in the Therapies and treatment section we provide a revision of both AB and Tau but also therapies targeted at APOE or the immune system.
Thirdly, there are issues with references. I think authors should spend some time checking the references.
We thank the reviewer for spotting that error. We have fixed the problem and reviewed the remaining references to ensure they are located appropriately.
Lastly, I would like you to talk about plant-based products used as potential targets for AD.
Some references that should be added:
- Alzheimer's disease: the silver tsunami of the 21 st century
https://www.ncbi.nlm.nih.gov/pmc/articles/PMC4904444/
- Exploring the efficacy of natural products in alleviating Alzheimer’s disease
P Deshpande, N Gogia, A Singh Neural regeneration research 14 (8), 1321
As per request of the reviewer, we have added these references within the “3. Early prediction and diagnosis are key to better treatment / Treatment” section of the reference as follows:
“natural products have the benefit of being readily available, as such some of them are being tested in animal models, for their neuroprotective, anti-inflammatory, antioxidant, anti-amyloidogenic, anticholinesterase properties, as potential therapeutics for AD [Sarkar et al. 2016; Deshpande et al. 2019,].”